# Neighbor-Aware Non-Orthogonal Multiple Access Scheme for Energy Harvesting Internet of Things

**DOI:** 10.3390/s22020448

**Published:** 2022-01-07

**Authors:** Yumi Kim, Mincheol Paik, Bokyeong Kim, Haneul Ko, Seung-Yeon Kim

**Affiliations:** Department of Computer and Information Science, Korea University, Sejong 30019, Korea; yumikim1201@korea.ac.kr (Y.K.); fall123123@korea.ac.kr (M.P.); pink9120@korea.ac.kr (B.K.); kimsy8011@korea.ac.kr (S.-Y.K.)

**Keywords:** game theory, constrained stochastic game, energy, energy harvesting, Internet of Things (IoT)

## Abstract

In a non-orthogonal multiple access (NOMA) environment, an Internet of Things (IoT) device achieves a high data rate by increasing its transmission power. However, excessively high transmission power can cause an energy outage of an IoT device and have a detrimental effect on the signal-to-interference-plus-noise ratio of neighbor IoT devices. In this paper, we propose a neighbor-aware NOMA scheme (NA-NOMA) where each IoT device determines whether to transmit data to the base station and the transmission power at each time epoch in a distributed manner with the consideration of its energy level and other devices’ transmission powers. To maximize the aggregated data rate of IoT devices while keeping an acceptable average energy outage probability, a constrained stochastic game model is formulated, and the solution of the model is obtained using a best response dynamics-based algorithm. Evaluation results show that NA-NOMA can increase the average data rate up to 22% compared with a probability-based scheme while providing a sufficiently low energy outage probability (e.g., 0.05).

## 1. Introduction

Internet of Things (IoT) devices have a limited battery capacity [1], which is one of the characteristic drawbacks of IoT systems. To overcome this drawback, many researchers have focused on the use of energy harvesting [2,3,4]. With energy harvesting, IoT devices can charge their battery by themselves, and therefore the operators of IoT systems do not need to replace and charge the IoT devices’ batteries, which provides low operating expenditure.

Meanwhile, many researchers have given much attention to non-orthogonal multiple access (NOMA) because it can significantly improve the spectral efficiency [5,6,7]. Additionally, the 5G standard defined by the third generation partnership project (3GPP) supports two types of NOMA: (1) power-domain-based NOMA and (2) code-domain-based NOMA. In the first type (i.e., power-domain-based NOMA), different powers are assigned to different IoT devices by considering their channel conditions for simultaneous transmissions to the base station (BS). On the other hand, BS can retrieve the desired signals using the received power difference and some signal processing techniques (e.g., successive interference cancellation). In the code-domain-based NOMA, different IoT devices use distinct codes, and therefore they are multiplexed over the same frequency. In general, the power-domain-based NOMA can be simply implemented without any significant changes of existing networks compared with the code-domain-based NOMA [7]. Specifically, for the power-domain-based NOMA, at the receiver side (i.e., BS), the only simple multiuser detection algorithm such as successive interference cancellation is needed. On the other hand, the transmitter (i.e., IoT devices) does not need any modification. Therefore, we consider the power-domain-based NOMA in this paper.

In the power-domain-based NOMA, each IoT device can increase its data rate by increasing its transmission power. However, excessively high transmission power can cause an energy outage of an IoT device and have a detrimental effect on the signal-to-interference-plus-noise ratio (SINR) of neighbor IoT devices, which indicates that each IoT device should choose the appropriate transmission power to maximize the aggregated data rate of IoT devices while preventing its energy outage. Even though a centralized approach can help decide transmission powers of IoT devices, a huge signaling overhead can occur and a centralized entity can be overloaded.

In this paper, we propose a neighbor-aware NOMA scheme (NA-NOMA) where each IoT device determines whether to transmit data to BS and the transmission power at each time epoch in a distributed manner by considering its energy level and other devices’ (i.e., neighbors’) transmission powers. To maximize the aggregated data rate of IoT devices while keeping an acceptable average energy outage probability, a constrained stochastic game model is formulated, and the solution of the model is obtained by means of a best response dynamics-based algorithm. Evaluation results show that NA-NOMA can increase the average data rate up to 22% compared to a probability-based scheme while providing a sufficiently low energy outage probability (e.g., 0.05). In addition, it is found that IoT devices using NA-NOMA adaptively change the operations (e.g., transmission powers) according to their environments (e.g., harvesting probability).

The contributions of this paper are as follows: (1) we optimize the policy on the transmission of IoT devices to maximize the aggregated data rate of IoT devices while keeping an acceptable average energy outage probability under the energy harvesting NOMA environment; (2) the optimal transmission policy can be achieved with small overhead (i.e., few iterations), and therefore the proposed scheme can easily achieve the optimal performance in practical systems; (3) we evaluate the proposed scheme under diverse environments, which can give useful guidelines to design energy harvesting and NOMA-based IoT systems.

For the remainder of this paper, we summarize related works in Section 2 and elaborate on NA-NOMA in Section 3. In Section 4, we formulate the constrained stochastic game model. In Section 5, we present evaluation results. Finally, the conclusion is given in Section 6.

## 2. Related Work

The objective of this paper is to maximize the aggregated data rate of IoT devices while keeping an acceptable average energy outage probability. There are many works with similar objectives (i.e., to increase the data rate and/or the energy efficiency) in the literature [8,9,10,11,12,13,14,15,16,17,18,19,20,21,22,23].

To maximize the data rate and energy efficiency, Huang et al. [8] devised an optimal power allocation algorithm using deep learning in multiple-input multiple-output (MIMO)-NOMA environments. Khan et al. [9] formulated an optimization problem to improve the aggregated data rate and reduce the energy consumption under some constraints such as QoS requirement and maximum transmission power. Cejudo et al. [10] developed an adaptive power allocation scheme that adaptively selects the combination of a suitable power allocation factor and modulation and coding scheme under the constraint of bit error rate. Liu and Petrova [11] formulated an optimization problem on the power allocation for the maximum aggregated data rate and designed a low complexity algorithm for achieving a near-optimal solution to the problem. Zhang et al. [12] proposed a two-step scheduling and power allocation scheme to optimize a tradeoff between consumed energy and achievable data rate. Fang et al. [13] formulated an optimization problem to optimize the overall energy efficiency under heterogeneous networks. In addition, they proposed a suboptimal algorithm based on mathematical techniques (i.e., relaxation and dual-decomposition). Zhang et al. [14] defined a resource optimization problem for NOMA with the consideration of heterogeneous networks, and designed a heuristic algorithm having low complexity by converting the problem into an equivalent subtractive one. Pei et al. [15] formulated a joint optimization problem on the power and time allocation for the maximized energy efficiency of device-to-device communications using NOMA. With similar decision variables, Cao and Zhao [16] formulated an optimization problem and divided the problem into two subproblems to develop a heuristics algorithm. Han et al. [17] investigated a channel and power allocation to optimize the energy efficiency of IoT devices by considering the characteristics of short-packet communication. Xu et al. [18] formulated a resource allocation problem in NOMA-based backscatter networks to maximize the energy efficiency of IoT devices while satisfying the QoS requirement. Zeng et al. [19] conducted an analytical study on the performance of NOMA, especially on the achievable data rate. Shen et al. [20] formulated a weighted aggregated data rate maximization problem and converted the formulated problem into a convex problem using mathematical techniques. Sreya et al. [21] proposed an adaptive rate NOMA where mobile users adaptively change their modulation and coding scheme (MCS) according to the channel conditions whereas IoT devices exploit the fixed MCS to improve the rate of successful transmissions of the IoT devices. Duan et al. [22] designed a resource allocation algorithm based on the K-means clustering method and matching theory to maximize the aggregated data rate. Na et al. [23] formulated a joint optimization problem on the trajectory of an unmanned aerial vehicle and the resource allocation to improve the average aggregated data rate. In addition, they developed a suboptimal iterative algorithm to solve the formulated problem.

However, there is no previous work to optimize the policy on the transmission of IoT devices under the energy harvesting and NOMA environment in a distributed manner.

## 3. Neighbor-Aware Non-Orthogonal Multiple Access Scheme

As shown in our system model (see Figure 1), all IoT devices have energy harvesting capabilities. That is, they harvest the energy from external sources (e.g., sun and wind) and store the energy in their battery with the capacity $E_{\max}$. In addition, IoT devices periodically generate data (e.g., senses temperature and monitors target) and determine whether to transmit these data to BS by consuming the energy stored in their battery. Because IoT devices exploit NOMA, each IoT device can increase its data rate by increasing its transmission power. However, excessively high transmission power can cause energy depletion and decrease the SINR of neighbor IoT devices. To sum up, it is important for each IoT device to decide whether to transmit data to BS and the transmission power to maximize the aggregated data rate of IoT devices while keeping a low energy outage probability. In addition, this decision should be made by considering the energy level and other devices’ (i.e., neighbors’) transmission powers. For this, a constrained stochastic game is formulated in the next section. Note that two types of decision (i.e., the decision on whether to transmit data to BS and the decision on the transmission power) can be handled by one dimension of action space, i.e., the transmission power, because the action representing zero transmission power can be interpreted as the situation where the IoT device does not transmit data to BS.

## 4. Constrained Stochastic Game

To maximize the aggregated data rate while keeping a low energy outage probability, we formulate a constrained stochastic game model [24,25]. It is assumed that IoT devices are the players, and therefore, in our formulation, player *i* and IoT device *i* have the same meaning. Table 1 summarizes the notations used in the model.

### 4.1. State Space

The local state space of IoT device *i*, $S_{i}$, can be defined as
(1)$$\begin{array}{c} S_{i}=\left\{0,1,2,...,E_{\max}\right\} \end{array}$$
where $E_{\max}$ denotes the maximum energy level (i.e., battery capacity) of IoT device *i*.

The global state space, S, can be represented as
(2)$$\begin{array}{c} S=\prod_{i}S_{i} \end{array}$$
where ∏ is the Cartesian product.

Meanwhile, let $S_{-i}$ describe the state space of all IoT devices except IoT device *i*.

### 4.2. Action Space

Because IoT device *i* should decide whether to transmit data and its transmission power, the local action space of IoT device *i*, $A_{i}$, can be represented as
(3)$$\begin{array}{c} A_{i}=\left\{0,1,\ldots ,A_{\max}\right\} \end{array}$$
where $A_{\max}$ denotes the maximum transmission power of IoT device *i*. $A_{i}$ (≠0) represents the transmission power of IoT device *i*. In addition, $A_{i}=0$ describes the situation where IoT device *i* does not conduct a transmission.

The global action space, A, can be described as
(4)$$\begin{array}{c} A=\prod_{i}A_{i} \end{array}$$

Similar to $S_{-i}$, the action space of all IoT devices except IoT device *i* is denoted as $A_{-i}$.

### 4.3. Transition Probability

$P[S_{i}^{\prime }|S_{i},A_{i}]$ denotes the probability of transiting a specific state $S_{i}$ to another state $S_{i}^{\prime }$ when IoT device *i* conducts the action $A_{i}$.

Even though IoT device *i* has an energy harvesting capability, it cannot always harvest energy because of the dynamicity of the energy source (e.g., wind and solar). That is, only when the energy source provides sufficient energy can IoT device *i* harvest energy. In this context, the harvesting process of IoT device *i* can be assumed as a Bernoulli random process that takes binary values (i.e., 0 and 1). That is, IoT device *i* can harvest a unit of energy with the probability $P_{i}^{H}$ [26]. Naturally, it cannot harvest any energy with the probability $1-P_{i}^{H}$. Thus, when IoT device *i* does not conduct a data transmission (i.e., $A_{i}=0$) and has room in its battery to store the harvested energy (i.e., $S_{i}\neq E_{\max}$), its energy $S_{i}$ can increase by a unit with probability $P_{i}^{H}$. On the other hand, IoT device *i* does not have any room in its battery (i.e., $S_{i}=E_{\max}$), its energy level does not change. To summarize, $P[S_{i}^{\prime }|S_{i}\neq E_{\max},A_{i}=0]$ and $P[S_{i}^{\prime }|S_{i}=E_{\max},A_{i}=0]$ can be represented as    
(5)$$\begin{array}{c} P\left[S_{i}^{\prime }|S_{i}\neq E_{\max},A_{i}=0\right]=\left\{\begin{array}{c} P_{i}^{H}\;\;\;\;\;\;\;\;\mathrm{if}\;S_{i}^{\prime }=S_{i}+1\\ 1-P_{i}^{H}\;\;\;\mathrm{if}\;S_{i}^{\prime }=S_{i}\\ 0,\;\;\;\;\;\;\;\;\;\;\;\mathrm{otherwise} \end{array}\right. \end{array}$$
and
(6)$$\begin{array}{c} P\left[S_{i}^{\prime }|S_{i}=E_{\max},A_{i}=0\right]=\left\{\begin{array}{c} 1,\;\mathrm{if}\;S_{i}^{\prime }=S_{i}\\ 0,\;\mathrm{otherwise}. \end{array}\right. \end{array}$$

When IoT device *i* transmits the data with power $A_{i}$ during one time epoch (i.e., a unit time), it is assumed that it consumes $A_{i}$ units of energy. When there is no sufficient energy in the battery of IoT device *i* (i.e., $S_{i}<A_{i}$), it does not transmit the data, which indicates that the energy of IoT device *i* does not decrease and can increase by a unit with probability $P_{i}^{H}$. Thus, $P[S_{i}^{\prime }|S_{i}\geq A_{i},A_{i}\neq 0]$ and $P[S_{i}^{\prime }|S_{i}<A_{i},A_{i}\neq 0]$ can be represented as
(7)$$\begin{array}{c} P\left[S_{i}^{\prime }|S_{i}\geq A_{i},A_{i}\neq 0\right]=\left\{\begin{array}{c} P_{i}^{H}\;\;\;\;\;\;\;\;\mathrm{if}\;S_{i}^{\prime }=S_{i}+1-A_{i}\\ 1-P_{i}^{H}\;\;\;\mathrm{if}\;S_{i}^{\prime }=S_{i}-A_{i}\\ 0,\;\;\;\;\;\;\;\;\;\;\;\mathrm{otherwise} \end{array}\right. \end{array}$$
and
(8)$$\begin{array}{c} P\left[S_{i}^{\prime }|S_{i}<A_{i},A_{i}\neq 0\right]=\left\{\begin{array}{c} P_{i}^{H}\;\;\;\;\;\;\;\;\mathrm{if}\;S_{i}^{\prime }=S_{i}+1\\ 1-P_{i}^{H}\;\;\;\mathrm{if}\;S_{i}^{\prime }=S_{i}\\ 0,\;\;\;\;\;\;\;\;\;\;\;\mathrm{otherwise}. \end{array}\right. \end{array}$$

### 4.4. Reward Function

The aggregated data rate of IoT devices is used as the reward function $r(S_{i},A_{i})$. Therefore, $r(S_{i},A_{i})$ can be defined as
(9)$$\begin{array}{c} r\left(S_{i},A_{i}\right)=\sum_{i}\log_{2}\left(1+\gamma_{i}\right) \end{array}$$
where $\gamma_{i}$ is the expected SINR of IoT device *i*.

$\gamma_{i}$ can be calculated by
(10)$$\begin{array}{c} \gamma_{i}=\frac{{\left|h_{i}\right|}^{2}A_{i}}{\sum_{i^{\prime }}{\left|h_{i^{\prime }}\right|}^{2}{\bar{A}}_{i^{\prime }}+\sigma^{2}} \end{array}$$
where $\bar{A_{i^{\prime }}}$ is the expected power of IoT device $i^{\prime }$. In addition, $|h_{i}{|}^{2}$ and $|h_{i^{\prime }}{|}^{2}$ are the channel gain of IoT device *i* and $i^{\prime }$, respectively. $\sigma^{2}$ is the noise power.

### 4.5. Constraint Function

For keeping a low energy outage probability, we define the constraint function $c\left(S_{i},A_{i}\right)$. The situation where there is no energy in the battery of IoT device *i* represents the energy outage. Therefore, the constraint function $c\left(S_{i},A_{i}\right)$ can be defined as
(11)$$\begin{array}{c} c\left(S_{i},A_{i}\right)=\left\{\begin{array}{c} 1,\;\mathrm{if}\;S_{i}=0\\ 0,\;\mathrm{otherwise}. \end{array}\right. \end{array}$$

### 4.6. Optimization Formulation

When π denotes a stationary policy of all players (i.e., IoT devices), a long-term average aggregated data rate $\zeta_{R}$ can be defined as
(12)$$\begin{array}{c} \zeta_{R}\left(\pi \right)=\lim_{T\to \infty }\frac{1}{T}\sum_{t=1}^{T}E_{\pi }\left[r\left(S^{t},A^{t}\right)\right] \end{array}$$
where $S^{t}$ and $A^{t}$ are the global state and the action at time *t*, respectively.

Meanwhile, IoT device *i* tries to maintain the long-term energy outage probability $\xi_{E}$ below a certain level, which can be described by
(13)$$\begin{array}{c} \xi_{E}\left(\pi \right)=\lim_{T\to \infty }\frac{1}{T}\sum_{t=1}^{T}E_{\pi }\left[c\left(S^{t},A^{t}\right)\right]\leq \theta_{E} \end{array}$$
where $\theta_{E}$ is the target energy outage probability.

Let $\pi_{i}^{*}$ and $\pi_{-i}^{*}$ denote the optimal policies (i.e., Nash equilibrium) (note that, if a stochastic game is formulated with a finite number of players, states, and actions, there is always Nash equilibrium of the game [27]) of IoT device *i* and all devices except IoT device *i*, respectively. Then, the integrated optimal policy (i.e., constrained Nash equilibrium) can be described by $\pi^{*}=\left(\pi_{i}^{*},\pi_{-i}^{*}\right)$. Note that, for any other stationary policy $\pi_{i}$ of IoT device *i*, $\zeta_{R}\left(\left(\pi_{i}^{*},\pi_{-i}^{*}\right)\right)\geq \zeta_{R}\left(\left(\pi_{i}^{},\pi_{-i}^{*}\right)\right)$ while satisfying the constraint. Meanwhile, when the policies of all IoT devices except IoT device *i* (i.e., $\pi_{-i}$) are given, the optimal policy $\pi_{i}^{*}$ of IoT device *i* (i.e., best response policy) should satisfy the inequality $\zeta_{R}\left(\left(\pi_{i}^{*},\pi_{-i}^{}\right)\right)\geq \zeta_{R}\left(\left(\pi_{i}^{},\pi_{-i}^{}\right)\right)$.

When $\phi_{i,\pi_{-i}}\left(S_{i},A_{i}\right)$ denotes the stationary probability that IoT device *i* chooses the action $A_{i}$ in local state $S_{i}$ with the given stationary policies of other IoT devices $\pi_{-i}$, we can formulate the equivalent LP model and its solution $\phi_{i,\pi_{-i}}^{*}\left(S_{i},A_{i}\right)$ can be interpreted as the optimal policy of the formulated game [28]. The LP model can be formulated as
(14)$$\begin{array}{c} \max_{\phi (S,A)}\sum_{S}\sum_{A}\phi_{i,\pi_{-i}}\left(S_{i},A_{i}\right)r\left(S_{i},A_{i}\right) \end{array}$$
(15)$$\begin{array}{c} s.t.\;\;\;\sum_{S}\sum_{A}\phi_{i,\pi_{-i}}\left(S_{i},A_{i}\right)c\left(S_{i},A_{i}\right)\leq \theta_{B} \end{array}$$
(16)$$\begin{array}{c} \sum_{A}\phi_{i,\pi_{-i}}\left(S_{i}^{\prime },A_{i}\right)=\sum_{S}\sum_{A}\phi_{i,\pi_{-i}}\left(S_{i},A_{i}\right)P\left[S_{i}^{\prime }|S_{i},A_{i}\right] \end{array}$$
(17)$$\begin{array}{c} \sum_{S}\sum_{A}\phi_{i,\pi_{-i}}\left({S_{i}}^{\prime },A_{i}\right)=1 \end{array}$$
(18)$$\begin{array}{c} \phi_{i,\pi_{-i}}\left({S_{i}}^{\prime },A_{i}\right)\geq 0 \end{array}$$

For maximizing the aggregated average data rate of IoT devices, we define the objective function in (Equation 14). The average energy outage probability can be maintained below the target energy outage probability $\theta_{E}$ by the constraint in (Equation 15). The Chapman–Kolmogorov equation is represented in the constraint in (Equation 16). The basic probability properties can be satisfied with the constraints in (Equation 17) and (Equation 18).

The optimal policy of IoT device *i* is given by
(19)$$\begin{array}{c} \pi_{i}^{*}\left(S_{i},A_{i}\right)=\frac{\phi_{i,\pi_{-i}}^{*}\left(S_{i},A_{i}\right)}{\sum_{A_{i}^{\prime }\in A_{i}}\phi_{i,\pi_{-i}}^{*}\left(S_{i},A_{i}^{\prime }\right)}. \end{array}$$

To achieve the optimal policies (i.e., best response policies) of IoT devices, we develop a best response dynamics-based algorithm as in Algorithm 1. First, the policy for all IoT devices is initialized (line 1 in Algorithm 1). After that, IoT devices share the channel gain with each other (line 2 in Algorithm 1). (We assume that the channel gain is stable until the algorithm is done. This assumption is reasonable because few iterations are needed to converge the optimal policies (see Section 5.1).) Meanwhile, each IoT device transmits the expected transmission power $\bar{A_{i}}$ to other IoT devices to interact with each other (line 5 in Algorithm 1). The expected transmission power of IoT device *i* can be obtained from $\bar{A_{i}}=\sum_{S_{i}}\sum_{A_{i}}A_{i}\phi_{i,\pi_{-i}}\left(S_{i},A_{i}\right)$. After receiving the expected transmission power from other IoT devices, each IoT device calculates the aggregated data rate of IoT devices and solves the LP problem to obtain the optimal policy $\pi_{i}^{*}$ (lines 6–7 in Algorithm 1). The algorithm is finished when the policies of all IoT devices converge.
**Algorithm 1:** Best response dynamics-based algorithm.  1:Initialize the policy $\pi_{i}$ for ∀i  2:Share the channel gain among IoT devices  3:**repeat**  4:**for** All IoT devices *i* **do**  5:Transmit the expected transmission power $\bar{P_{i}}$ to other IoT devices  6:Calculate the aggregated data rate of IoT devices  7:Solve the LP problem to obtain the optimal policy $\pi_{i}^{*}$  8:**end for**  9:**until** Convergence of the policies for all IoT devices

Because the complexity of solving the LP problem is not high (e.g., $O((|S_{i}\left|\right|A_{i}{\left|\right)}^{3})$ [29] for the Vaidya’s algorithm which is one of the representative LP solvers) and the stationary policies can be achieved with few iterations (see Section 5.1), the proposed algorithm can be realized without a huge overhead.

## 5. Evaluation Results

For the performance evaluation, we introduce the following comparison schemes: (1) MAX where IoT devices always transmit data with the maximum power; (2) MIN where IoT devices always conduct the transmission with the minimum power; (3) RAND where IoT devices randomly select their actions; (4) PROB where IoT devices select their actions according to the predefined probability. (In PROB, A=0, A=0, A=0, A=0, and A=0 are selected by 0.1, 0.3, 0.25, 0.25, and 0.1. Even though we have conducted extensive simulations with various probabilities, we cannot obtain different results. Therefore, in this paper, only results with the mentioned probabilities are included.) The performance metrics are the aggregated data rate of IoT devices $\zeta_{R}$ and the energy outage probability $\xi_{E}$.

We have conducted the simulation with the following default parameters. The total number of IoT devices is 5. The maximum energy level of IoT devices, $E_{\max}$, is 9. In addition, the maximum transmission power $A_{\max}$ is 4. The energy harvesting probability $P^{H}$ of IoT devices is set to [0.30.7], where [ab] denotes a random value between *a* and *b*. In addition, we consider that each device has i.i.d. Rayleigh fading channel and its mean is set to [02] [30]. $\sigma^{2}$ is set to 0.5 [30]. The target energy outage probability $\theta_{E}$ is set to 0.05.

### 5.1. Convergence to Nash Equilibrium

Figure 2 describes a process that the policies of IoT devices converge to the integrated optimal policy (i.e., Nash equilibrium). As illustrated in Figure 2, the proposed algorithm (i.e., Algorithm 1) can find the integrated optimal policy within a few iterations (i.e., 1) (note that the initial policy of each IoT device is randomly set). Specifically, the initial policy on the transmission of IoT devices is randomly set (see 0 iterations in Figure 2). Then, based on the proposed algorithm, each IoT device finds the optimal policy (i.e., Nash equilibrium) only after one iteration. After finding the optimal policy, each IoT device does not need to change its policy, and therefore the policies of the IoT devices do not change after the first iteration. This result implies that our algorithm can be realized in the energy harvesting NOMA environment without high signaling overhead.

Meanwhile, from Figure 2, it can be found that each IoT device chooses its action by considering the other devices’ actions at the best response. Specifically, after the convergence, because IoT device 1 chooses the maximum transmission power (i.e., A=4) with low probability, other IoT devices choose the maximum transmission power with a higher probability to improve the expected SINR.

### 5.2. Effect of $P^{H}$

Figure 3 shows the change of the average aggregated data rate $\zeta_{R}$ and the energy outage probability $\xi_{E}$ according to the energy harvesting probability $P^{H}$. From Figure 3, we can observe that NA-NOMA can achieve the highest average aggregated data rate $\zeta_{R}$ while keeping the acceptable energy outage probability (i.e., 0.05). This is because IoT devices in NA-NOMA determine whether to transmit data to BS and their transmission powers by considering the energy level and other devices’ actions (i.e., transmission powers). For example, a specific IoT device can transmit its data with high transmission power to increase the average aggregated data rate, when its energy level is high and other IoT devices transmit data with low transmission power (or do not transmit data). Moreover, if an IoT device has a low energy level, it does not transmit data to avoid an energy outage.

Meanwhile, from Figure 3a, it can be observed that $\zeta_{R}$ of NA-NOMA increases as $P^{H}$ increases. This is explained in the following. High $P^{H}$ indicates that IoT devices can transmit data without any concern about the energy outage. In this situation, IoT devices in NA-NOMA aggressively exploit high transmission power. Meanwhile, because the other schemes do not adjust their transmission policies by considering the energy harvesting probability, their average aggregated data rate is constant regardless of $P^{H}$ (see Figure 3a), and their energy outage probabilities decrease as $P^{H}$ increases (see Figure 3b).

Meanwhile, as shown in Figure 3a, the average aggregated data rates of the comparison schemes are similar to each other. This is because all IoT devices in each comparison scheme operate with the same policy, which can degrade the expected SINR. For example, even if a specific IoT device transmits its data with the maximum transmission power, the other IoT devices probably transmit their data with the maximum transmission power, which causes low expected SINR.

### 5.3. Effect of $\theta_{E}$

Figure 4 shows the effect of the target energy outage probability $\theta_{E}$ on the average aggregated data rate $\zeta_{R}$ and the energy outage probability $\xi_{E}$. As observed in Figure 4a, $\zeta_{R}$ of NA-NOMA increases as $\theta_{E}$ increases. This is because higher $\theta_{E}$ means that IoT devices can try to transmit data with higher transmission power without less concern about the energy outage. However, because the other comparison schemes operate based on the fixed policy without consideration of the target energy outage probability $\theta_{E}$, their average aggregated data rate and the energy outage probability do not change regardless of $\theta_{E}$ (see Figure 4a,b).

## 6. Conclusions

This paper proposes NA-NOMA where each IoT device considers its energy level and other devices’ transmission powers to decide its transmission policy. For example, each IoT device can transmit data with high transmission power when its energy level is sufficient and/or other devices do not transmit any data. In addition, if the energy level of an IoT device is not sufficient and/or there are lots of other devices transmitting data, that IoT device does not transmit any data. To maximize the aggregated data rate of IoT devices while keeping the acceptable average energy outage probability, a constrained stochastic game model is formulated. To achieve the optimal (i.e., best response) policies of IoT devices, the best response dynamics-based algorithm with the polynomial complexity is introduced. Evaluation results demonstrate that the proposed algorithm can find the optimal stationary policies within a few iterations. In addition, it can be shown that NA-NOMA can increase the average data rate up to 22% compared with a probability-based scheme while providing a sufficiently low energy outage probability (e.g., 0.05). Moreover, it can be observed that NA-NOMA optimizes its transmission policy by considering its operating environment (e.g., energy harvesting probability and target energy outage probability). As one of future works, we will investigate the spatiotemporal correlation of energy harvesting probabilities of IoT devices to further reduce the energy outage probability. In addition, we will develop the prototype of the proposed scheme for IoT-based environmental monitoring systems.

## Figures and Tables

**Figure 1 sensors-22-00448-f001:**
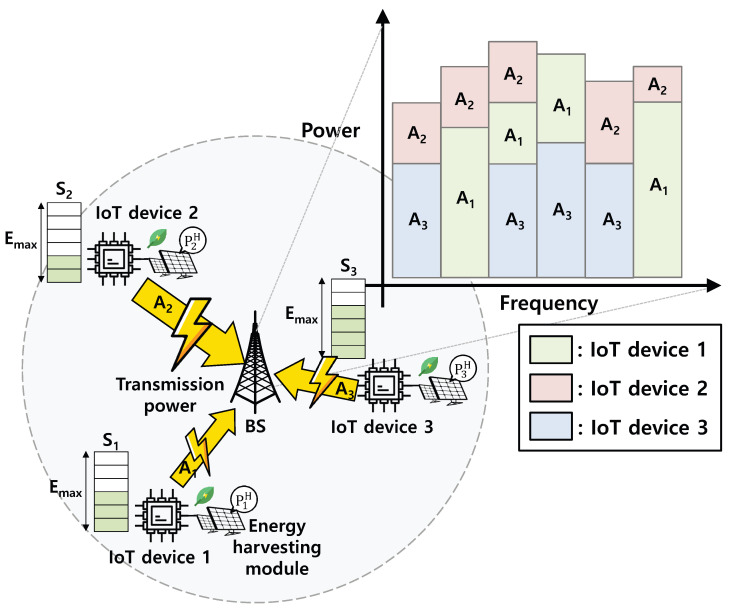
System model.

**Figure 2 sensors-22-00448-f002:**
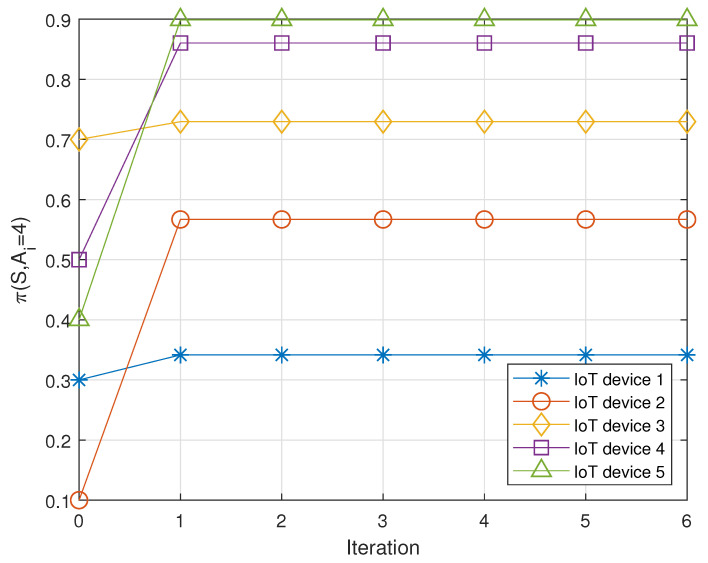
Process that the policies of IoT devices converge to the integrated optimal policy (i.e., Nash equilibrium).

**Figure 3 sensors-22-00448-f003:**
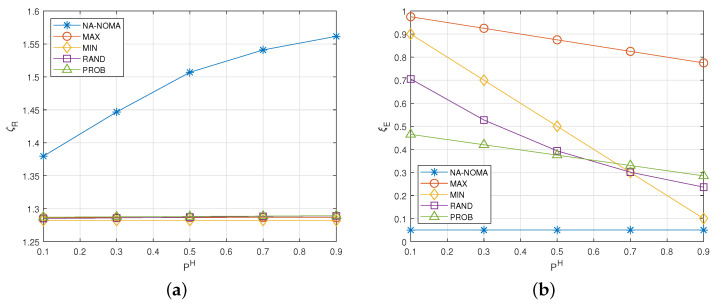
Effect of $P^{H}$. (**a**) Average aggregated data rate. (**b**) Energy outage probability.

**Figure 4 sensors-22-00448-f004:**
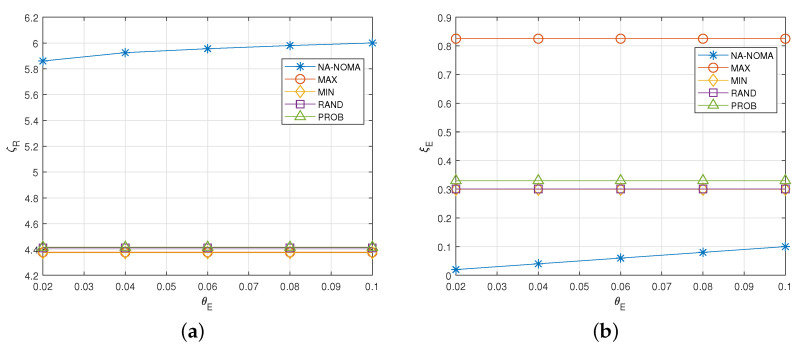
Effect of $\theta_{E}$. (**a**) Average aggregated data rate. (**b**) Energy outage probability.

**Table 1 sensors-22-00448-t001:** Summary of notations.

Notation	Description
$S_{i}$	Local state space of IoT device *i*
$S_{-i}$	State space of all IoT devices except IoT device *i*
S	Global state space
$E_{\max}$	Maximum energy level of IoT device *i*
$A_{i}$	Local action space of IoT device *i*
A	Global action space
$A_{\max}$	Maximum transmission power of IoT device *i*
$P_{i}^{H}$	Probability that IoT device *i* harvests one-unit of energy
$\gamma_{i}$	Expected SINR of IoT device *i*
$\bar{A_{i}}$	Expected power of IoT device *i*
$|h_{i}{|}^{2}$	Channel gain of IoT device *i*
$\sigma^{2}$	Noise power
$\zeta_{C}$	Average aggregated data rate of IoT devices
$\xi_{E}$	Average energy outage probability

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
