# Peer review of "Neighbor-Aware Non-Orthogonal Multiple Access Scheme for Energy Harvesting Internet of Things"

_sensors, 2022, doi:10.3390/s22020448_

Round 1

Reviewer 1 Report

From the article title point of view, the paper is well intentioned into energy harvesting topic. Congrats to the authors for aiming this research. 
State of the art is entirely relating to the current concern of the nowadays solution and the research interest area. From referances, there are more than 80 percent from recent researches.
 At chapter 2- the paper does not present any information that must alocate an chapter. It can be asociatetd to the chapter 1 Introduction.
 At chapter 3 the figure 1 is not clearly presented. The system must be described more explicitly, it is not understanding where are used the  notations centralized into the table 1?!? The table format is not as template…
  At chapter 4 the reference [20] does not offer sufficient basic information related to Stochastic Game proposed….  It cannot be used a table in a previous chapter and used in a different chapter… please move the table in chapter 4….

Fig. 2 is not clear for what was introduced in article?!? What is the diference between 6 iteration ?! No changes are highlight… .
Why Algorithm 1? Is the any other algorithms ???
Conclusions are not complete…. Please improve them….  
The paper is interesting and can be published in the journal with some major adjustments.

Author Response

We really appreciate your time and valuable comments. We have carefully revised our paper taking into consideration of your comments and suggestions. Please find the attached response letter.

Reviewer 2 Report

In this work, the authors optimize the policy on the transmission of IoT devices to maximize the aggregated IoT’s data rate, while keeping an acceptable average energy outage probability under the energy harvesting NOMA environment. To do so, they formulated a constrained stochastic game model, before resolving it using a best response dynamics-based algorithm. The experimental results show that the proposed scheme can increase the average data rate up to 22% compared to a probability-based scheme, while providing a sufficiently low energy outage probability. I find this work very timely and addressing a very hot topic. The paper is well written and organized. Some comments that can be considered before a final publication:

  • The related work section should be extended to include more works, such as:
    • Sreya, S. Saigadha, P. D. Mankar, G. Das and H. S. Dhillon, "Adaptive Rate NOMA for Cellular IoT Networks," in IEEE Wireless Communications Letters, doi: 10.1109/LWC.2021.3132932.
    • Brik, M. Esseghir, L. Merghem-Boulahia and H. Snoussi, "ThingsGame: when sending data rate depends on the data usefulness in IoT networks," 2018 14th International Wireless Communications & Mobile Computing Conference (IWCMC), 2018, pp. 886-891, doi: 10.1109/IWCMC.2018.8450391.
  • It is not clear for me why the authors did not give any proof on the existence of Nash equilibrium.
  • Does this work suitable for the 5G standards such as 3GPP ?

Author Response

(The authors gave the same response as above.)

Round 2

Reviewer 1 Report

The authors have replied to all the queries and concerns with proper references. However, to improve the connectivity for readers, authors can develop their research in future articles by including some practical applications.

Good luck with future researches!

Author Response

We really appreciate your time and valuable comment. We have conducted the second round of revision based on your comment. Please find the attached file for the details.
